# Predicting Complications following Surgical Resection of Hepatocellular Carcinoma Using Newly Developed Neo-Glasgow Prognostic Score with ALBI Grade: Comparison of Open and Laparoscopic Surgery Cases

**DOI:** 10.3390/cancers14061402

**Published:** 2022-03-09

**Authors:** Masaki Kaibori, Atsushi Hiraoka, Kosuke Matsui, Hideyuki Matsushima, Hisashi Kosaka, Hidekazu Yamamoto, Takashi Yamaguchi, Katsunori Yoshida, Mitsugu Sekimoto

**Affiliations:** 1Department of Surgery, Kansai Medical University, Osaka 573-1191, Japan; matsuik@hirakata.kmu.ac.jp (K.M.); h.matsushima0921@gmail.com (H.M.); kosakahi@hirakata.kmu.ac.jp (H.K.); yamhidek@hirakata.kmu.ac.jp (H.Y.); sekimotm@hirakata.kmu.ac.jp (M.S.); 2Gastroenterology Center, Ehime Prefectural Central Hospital, Matsuyama 790-0024, Japan; hirage@m.ehime-u.ac.jp; 3Department of Gastroenterology and Hepatology, Kansai Medical University, Osaka 573-1191, Japan; yamaguct@hirakata.kmu.ac.jp (T.Y.); yoshidka@takii.kmu.ac.jp (K.Y.)

**Keywords:** hepatocellular carcinoma, ALBI grade, Glasgow prognostic score, complication, neo-Glasgow prognostic score

## Abstract

**Simple Summary:**

Glasgow prognostic score (GPS), using with CRP and albumin has been shown to be an important and useful nutritional assessment tool for predicting prognosis in such cases. We developed a modified GPS scoring method (neo-GPS), based on ALBI grade and reported to indicate the approximate borderline of amino acid imbalance instead of serum albumin, in the present study. The present results indicate that neo-GPS has not only better predictive value for prognosis but also shows greater sensitivity for predicting risk of postoperative complications as compared to GPS in patients undergoing a hepatectomy for hepatocellular carcinoma.

**Abstract:**

Background/Aim: Nutritional assessment is known to be important for predicting prognosis in patients with malignant diseases. This study examined the usefulness of a prognostic predictive nutritional assessment tool for hepatocellular carcinoma (HCC) patients treated with surgical resection. Materials/Methods: HCC patients (*n* = 429) classified as Child–Pugh A who underwent an R0 resection between 2010 and 2020 were retrospectively analyzed (median age 73 years, males 326 (76.0%), Child–Pugh score 5:6 = 326:103, single tumor 340 (79.2%), median tumor size 3.5 cm, open:laparoscopic = 304:125). Glasgow prognostic score (GPS) and the newly developed neo-GPS method, which uses albumin–bilirubin grade 1 instead of albumin, were evaluated to compare their usefulness for prognosis prediction. Results: Median survival time for patients with a GPS score of 0, 1, and 2 was 120, 51, and 20 months, respectively. As for neo-GPS, that for those with a score of 0, 1, and 2 was not applicable (NA), 53 months, and 35 months, respectively (each *p* < 0.001; c-index: 0.556 and 0.611, respectively). Furthermore, median progression-free survival was 33, 22, and 9 months, and 41, 24, and 15 months, respectively (each *p* < 0.001; c-index: 0.539 and 0.578, respectively). As compared to patients with a high GPS (≥1), those with a high neo-GPS (≥1) showed a greater rate of high Clavien–Dindo classification (≥3) (39.2% vs. 65.1%). A comparison of patients with a high GPS (≥1) with those with a high neo-GPS (≥1) showed no significant difference regarding frequency of open or laparoscopic hepatectomy (17.4% vs. 15.2%, *p* = 0.670; 44.7% vs. 43.2%, *p* = 0.831, respectively), while the frequency of high Clavien–Dindo classification (≥3) was lower in patients who underwent a laparoscopic hepatectomy (11.2% vs. 22.7%, *p* = 0.007). Conclusion: The present findings suggest that the newly developed neo-GPS based on ALBI grade is an effective prognostic nutritional assessment tool and can be used for prediction of postoperative complications.

## 1. Introduction

Hepatocellular carcinoma (HCC) is known to be the most common primary malignancy of the liver, while worldwide it is the fifth most common of all malignancies [1]. An open hepatectomy is the standard surgical treatment conventionally performed for HCC patients, though recently a laparoscopic hepatectomy method has been developed as a less invasive yet equal or more effective therapeutic modality. Along with advancements in techniques and instruments used for a laparoscopy over more than two decades, a laparoscopic hepatectomy has been increasingly adopted worldwide [2,3], as it is associated with a lower volume of intraoperative bleeding and fewer postoperative complications as compared with a conventional open liver resection, and now considered to be safe and feasible treatment modality for liver tumors.

Nutritional assessment is known to be important for predicting prognosis in patients with malignant diseases including HCC. Glasgow prognostic score (GPS) [4], defined based on C-reactive protein (CRP) (1.0 mg/dL) and serum albumin (3.5 g/dL) levels, has been shown to be an important and useful nutritional assessment tool for predicting prognosis in such cases [5,6,7]. In addition, as compared to patients with a low GPS score (<1), those with a score ≥1 have been reported to have a higher rate of complications following surgical resection for advanced gastric cancer (16% vs. 31%, *p* = 0.022) [8]. On the other hand, because HCC often develops in patients with chronic liver disease, mainly with liver cirrhosis, development of a new nutritional evaluation method, which fits for chronic liver disease patients, is needed.

In the present study, an improved GPS scoring method (neo-GPS) using albumin–bilirubin (ALBI) grade 1 reported to indicate the approximate borderline of amino acid imbalance [9] instead of serum albumin, was developed and a prognostic predictive value of newly developed neo-GPS was evaluated in comparison with GPS score following surgical resection.

## 2. Materials and Methods

The records of all HCC patients who underwent a liver resection procedure from January 2010 to September 2020 at Kansai Medical University Hospital (Osaka, Japan) were screened. A total of 484 patients with HCC underwent an R0 resection, defined as macroscopic removal of all tumors, of which 429 classified as Child–Pugh A were enrolled in the present study. All the procedures cited in this study were performed by only one surgeon, who has carried out over 1500 liver resection procedures. The Kansai Medical University’s institutional ethics committee approved the study protocol (approval number: KMU 2021311).

### 2.1. Underlying Liver Disease

Positive anti-HCV results indicated HCC caused by the hepatitis C virus (HCV). On the other hand, positive HBV surface antigen indicated HCC caused by the hepatitis B virus (HBV). For individuals with a history of alcohol abuse (≥60 g/day), the underlying liver disease was attributable to alcohol [10,11].

### 2.2. Liver Function and Nutritional Status Assessments

Child–Pugh score/classification [12] and ALBI grade [13,14] were used for hepatic reserve function assessment. Nutritional status was assessed using GPS [4] and neo-GPS, with ALBI grade 1 instead of serum albumin used in the latter. High GPS and neo-GPS were both defined as ≥1. 

### 2.3. Clinicopathologic Variables, HCC Treatment Algorithm, and Surgical Procedures 

Prior to surgery, patients were tested for the indocyanine green retention rate at 15 min (ICG-R15), and standard liver function tests were performed. Patients’ alpha-fetoprotein (AFP) levels were also measured. We employed a revised HCC therapy algorithm that considered five factors: liver function reserve, extrahepatic metastases, vascular invasion, tumor number, and tumor size [15]. The degree of liver damage (including an ICG-R15 measurement) was utilized to make a decision when hepatectomy was being considered. The revised treatment algorithm is summarized as follows. One of three treatment regimes is advised for HCC patients with Child–Pugh class A/B liver function with no extrahepatic metastasis or vascular invasion. First, for up to three HCCs measuring 3 cm or less, either surgical resection or radiofrequency ablation is indicated with no priority; otherwise, surgical resection is advised as first-line therapy for solitary HCCs regardless of size. Second, surgical resection is suggested as first-line therapy, and transarterial chemoembolization as second-line therapy for up to three HCCs measuring >3 cm. Third, a combination of embolization, hepatectomy, hepatic arterial infusion chemotherapy, and molecular targeted therapy is indicated for patients with HCC accompanied by vascular invasion but no extrahepatic metastases. Treatment is determined for each patient based on their unique circumstances, which includes parameters such as liver function, HCC status, and the degree of vascular invasion. Surgical operations were classified using the Brisbane terminology as proposed by Strasberg et al. [16]. Resection of the tumor, accompanying portal vein branches, and matching hepatic region, was referred to as anatomic resection. Hemihepatectomy (resection of half of the liver), extended hemihepatectomy (hemihepatectomy plus removal of additional contiguous segments), sectionectomy (resection of two Couinaud subsegments [17]), and segmentectomy (resection of two Couinaud subsegments) were the four types of anatomic resection. Limited resections were assigned to all other non-anatomic surgeries. Both peripheral and central tumors were treated using limited resection. Partial hepatectomy was utilized to treat peripheral tumors and extrahepatic proliferation because it allows for appropriate surgical margins. Enucleation, on the other hand, was utilized to treat central tumors near the hepatic hilum or major vessels because of the difficulty and risks associated with achieving appropriate margins. A senior pathologist analyzed each specimen and performed a histological examination to confirm the final diagnosis.

### 2.4. Evaluation of Complications following Surgical Resection

For evaluation of complications associated with surgical resection, the Clavien–Dindo classification [18] was used, with grade ≥3 considered to be a significant complication in the present study.

### 2.5. Statistical Analysis

Welch’s *t*-test, Mann–Whitney U test, Kaplan–Meier method, and a log-rank test were employed for the statistical analysis. Statistical significance was defined as a *p* value of less than 0.05. The Akaike information criterion (AIC) [19] was used to analyze the scoring models’ discriminatory capacities, and the c-index was employed to assess their predictive capabilities for prognosis. All statistical analyses were carried out using Easy-R (EZR), version 1.53 (Saitama Medical Center, Jichi Medical University, Saitama, Japan) [20], a graphical user interface for R (The R Foundation for Statistical Computing, Vienna, Austria).

## 3. Results

The median age of the present cohort was 73 years and 326 (76.0%) were male. An open hepatectomy was performed in 304 (70.9%) and a laparoscopic hepatectomy in 125 patients (29.1%) (Table 1).

Median survival time (MST) was 120 months (95%CI: 100–not applicable (NA)) in patients with GPS 0, 51 months (95%CI: 33–NA) in those with GPS 1, and 20 months (95%CI: 5–NA) in those with GPS 2 (*p* < 0.001) (Figure 1A), while that in patients with neo-GPS 0 was NA (95%CI: 114–NA), with neo-GPS 1 was 53 months (95%CI: 42–111), and with neo-GPS 2 was 35 months (95%CI: 16–NA) (*p* < 0.001) (Figure 1B). In a comparison of overall survival (OS), AIC was lower (1554 vs. 1562) and c-index higher (0.611 vs. 0.556) according to neo-GPS as compared with GPS.

The median progression-free survival (mPFS) period was 33 months (95%CI: 26–44) in patients with GPS 0, 22 months (95%CI: 12–25) in those with GPS 1, and nine months (95%CI: 3–NA) in those with GPS 2 (*p* < 0.001) (Figure 2A), while in that patients with neo-GPS 0 was 41 months (95%CI: 30–53), in those with neo-GPS 1 was 24 months (95%CI: 16–29), and in those with neo-GPS 2 was 15 months (95%CI: 9–24) (*p* < 0.001) (Figure 2B). Again, AIC was lower (2758 vs. 2765) and c-index was higher (0.578 vs. 0.539) based on neo-GPS compared to GPS.

Platelet count, total bilirubin, prothrombin time, and ICG-R15 results were worse in patients who received a laparoscopic hepatectomy as compared to those who underwent an open hepatectomy, while the open hepatectomy group showed a lower frequency of a single tumor and larger tumor size. Blood loss during the operation in the laparoscopic hepatectomy group was significantly lower than that in the open hepatectomy group (*p* < 0.001) (Table 2).

There were no significant differences for frequency of patients with a high GPS (≥1) and high neo-GPS (≥1) value between open hepatectomy and laparoscopic hepatectomy procedures (17.4% vs. 15.2%, *p* = 0.670; 44.7% vs. 43.2%, *p* = 0.831, respectively). Those that underwent a laparoscopic hepatectomy showed a lower rate of high Clavien–Dindo classification (≥3) (11.2% vs. 22.7%, *p* = 0.007), while patients with a high neo-GPS value (≥1) showed a greater rate of high Clavien–Dindo classification (≥3) as compared to those with a high GPS value (≥1) (65.1% vs. 32.5%) (Table 3). In sub-analysis findings of patients with a single tumor, the frequency of high Clavien–Dindo classification (≥3) was greater in those that underwent an open hepatectomy as compared to a laparoscopic hepatectomy procedure (22.0% (50/227) vs. 10.6% (12/113), *p* = 0.011). 

There was no significant difference for MST between the open (120 months, 95%CI: 100–NA) and laparoscopic (75 months, 95%CI: 52–NA) (*p* = 0.66) hepatectomy procedures (Figure 3A), or for progression survival (PFS) between those groups (28 months (95%CI: 22–36) vs. 30 months (95%CI: 21–42), *p* = 0.577) (Figure 3B). 

There were no significant differences in regard to (A) median overall survival between open (120 months, 95%CI: 100-not applicable (NA)) and laparoscopic hepatectomy (75 months, 95%CI: 52–NA) cases (*p* = 0.660), or for (B) progression-free survival (28 months (95%CI: 22–36) vs. 30 months (95%CI: 21–42), *p* = 0.577). 

## 4. Discussion

In the present study, the modified nutritional assessment tool neo-GPS was found to be a better method for predicting prognosis as compared to GPS, not only for OS but also PFS. While there were no significant differences for frequency of high GPS (≥1) and high neo-GPS (≥1) between patients treated with an open hepatectomy (17.4% vs. 15.2%, *p* = 0.670) and those with a laparoscopic hepatectomy (44.7% vs. 43.2%, *p* = 0.831), the latter group showed a lower rate of high Clavien–Dindo classification (≥3) (11.2% vs. 22.7%, *p* = 0.007). Additionally, it was noted that patients with a high neo-GPS showed a higher rate of high Clavien–Dindo classification (≥3) as compared to those with a high GPS (65.1% vs. 32.9%) (Table 3).

The results of the current study, namely, the lower rate of complications and less intraoperative bleeding volume associated with a laparoscopic hepatectomy as compared with an open hepatectomy (Table 2) confirm previously reported findings [21,22,23,24,25,26,27,28,29]. There are various possible reasons to explain the favorable short-term outcomes of a laparoscopic hepatectomy shown in this study. Although operation time between the two methods was not significantly different, use of a laparoscopy eliminates the need for an extensive adhesiolysis procedure [30,31], which may explain the shorter operative time and lower intraoperative bleeding volume as compared with an open hepatectomy. Furthermore, use of a laparoscopic hepatectomy potentially reduces occurrence of ascites [32,33]. This might be explained by minimal disruption of collateral circulation in the abdominal wall and lymphatic flow in the diaphragm in cases with portal hypertension as compared to an open hepatectomy, as a laparoscopic hepatectomy requires only four or five trocars in the upper quadrant of the abdomen instead of a large subcostal incision [34]. Additionally, with reduced postoperative pain and earlier postoperative weaning, a laparoscopic approach may reduce pulmonary complications, such as respiratory infections, pleural effusion, and respiratory failure [35]. As a result, a meticulous maneuver is made possible to reduce some risk of possible serious complications, including bile leakage, massive bleeding, intestinal damage, and liver failure [2,28,32,33,34,36].

A newly developed assessment tool for hepatic function, ALBI score/grade, was recently proposed [13] and has been shown to have a good relationship with ICG-R15 (r = 0.563, 95%CI: 0.550–0.570, *p* < 0.0001) [37]. In addition, ALBI score was found to have a good relationship with Onodera’s prognostic nutritional index (r = −0.939, 95%CI: −0.95 to −0.92, *p* < 0.001), used as a nutritional assessment tool in relation to prognosis [38]. HCC is well known to develop frequently in patients with chronic liver disease, who are often complicated with amino acid imbalance. Furthermore, it has been recommended that nutritional intervention be considered for cases of chronic liver disease with a serum albumin level ≤3.5 g/dL [39,40]. Additionally, it was recently reported that the borderline of amino acid imbalance shown by an ALBI score of −2.588 is similar to the cut-off value for ALBI grade 1 (−2.600) [9]. With the above factors in mind, neo-GPS was developed using ALBI grade instead of serum albumin (3.5 g/dL), as that is considered to be a more sensitive nutritional marker. In fact, although the cut-off value of albumin in GPS is 3.5 g/dL, even the exemplar patient with a very low total bilirubin level of 0.3 mg/dL and an albumin level of 3.5 g/dL is categorized as the middle grade of ALBI (grade 2, score −2.51). On the other hand, it has been reported that ALBI grade 1 is superior to Child–Pugh A in prediction of prognosis of HCC patients [14]. These facts suggest that the cut-off value of albumin of GPS (3.5 g/dL) is an unsatisfactory criterion for HCC patients whose basal diseases are often liver cirrhosis, and that ALBI grade should be used instead of albumin (3.5 g/dL) as a nutritional factor. Compared to GPS, which was developed as a nutritional assessment method for patients with cancers other than HCC, the present newly developed neo-GPS using ALBI grade 1 might play a greater role for predicting prognosis and fit for HCC patients with chronic liver disease than GPS, because ALBI grade 1 can indirectly assess the nutritional status for the boundaries of amino acid imbalance in patients with chronic liver disease [9].

The present results indicate that neo-GPS has not only better predictive value for prognosis but also shows greater sensitivity for predicting risk of postoperative complications as compared to GPS in patients undergoing a hepatectomy for hepatocellular carcinoma. Additionally, a laparoscopic hepatectomy might be safer than an open hepatectomy procedure in patients with high neo-GPS (≥1).

This study is limited by its retrospective nature. Furthermore, tumor number and size were greater in the open hepatectomy as compared to the laparoscopic hepatectomy cases. Because of lacking data (e.g., neutrophils and lymphocyte), we could not perform comparison among the present neo-GPS and neutrophil/lymphocyte ratio or nutritional index, therefore, future study to compare among the present neo-GPS and other nutritional assessment tools should be planned. Moreover, it is considered that a randomized control trial for obtaining more concrete conclusions is necessary in addition to validation studies, while accumulation of greater numbers of patients and a longer observation period will be useful for presenting more definitive conclusions.

We concluded that newly developed neo-GPS based on ALBI grade is a good prognostic nutritional assessment tool for prediction of postoperative complications.

## Figures and Tables

**Figure 1 cancers-14-01402-f001:**
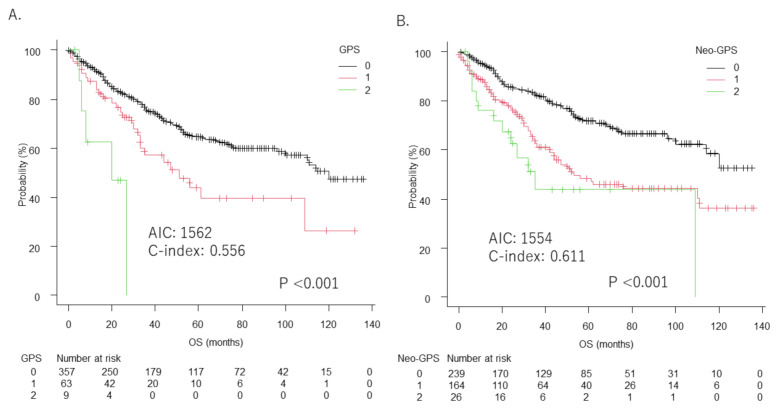
Comparison between GPS and neo-GPS for prediction of overall survival (OS). (**A**) Median overall survival (MST) for patients with GPS 0 was 120 months (95%CI: 100–NA), for those with GPS 1 was 51 months (95%CI: 33–NA), and for those with GPS 2 was 20 months (95%CI: 5–NA). (**B**) MST in patients with neo-GPS 0 was not applicable (NA) (95%CI: 114–NA), for those with neo-GPS 1 was 53 months (95%CI: 42–111), and for those with neo-GPS 2 was 35 months (95%CI: 16–NA) (*p* < 0.001) (**B**). The AIC and c-index values for GPS were 1562 and 0.556, respectively, and for neo-GPS were 1554 and 0.611, respectively.

**Figure 2 cancers-14-01402-f002:**
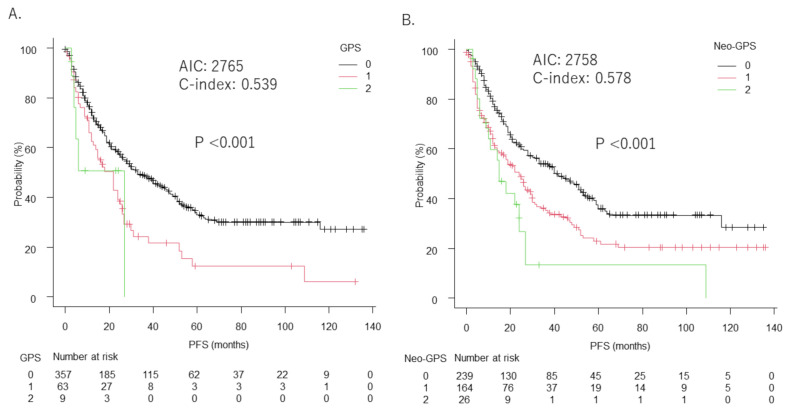
Comparison between GPS and neo-GPS for prediction of progression-free survival (PFS). (**A**) Median progression survival (mPFS) for patients with GPS 0 was 33 months (95%CI: 26–44), for those with GPS 1 was 22 months (95%CI: 12–25), and for those with GPS 2 was 9 months (95%CI: 3–NA). (**B**) mPFS for patients with neo-GPS 0 was 41 months (95%CI: 30–53), for those with neo-GPS 1 was 24 months (95%CI: 16–29), and for those with neo-GPS 2 was 15 months (95%CI: 9–24) (*p* < 0.001). AIC and c-index for GPS were 2765 and 0.539, respectively, and for neo-GPS were 2758 and 0.578, respectively.

**Figure 3 cancers-14-01402-f003:**
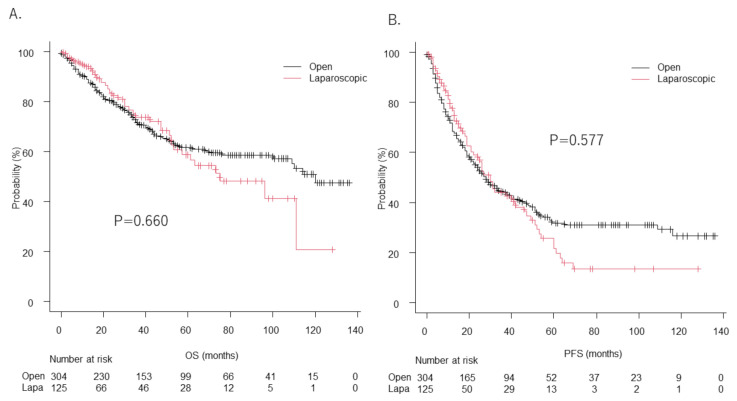
Overall survival (OS) (**A**) and progression-free survival (PFS) (**B**) in open and laparoscopic hepatectomy cases.

**Table 1 cancers-14-01402-t001:** Clinical features of patients (*n* = 429).

Age, years *	73 (66–78)
Gender, male:female	326:103
Body mass index, kg/m^2^ *	23.2 (20.9–25.8)
Etiology, HCV:HBV:HBV&HCV:alcohol:others	154:74:5:53:143
Positive for diabetes mellitus, *n* (%)	141 (32.9)
Aspartate transaminase, U/L *	34 (25–49)
Alanine aminotransferase, U/L *	28 (18–46)
Platelets, 10^4^/µL *	15.7 (11.9–20.7)
Total bilirubin, mg/dL *	0.7 (0.6–1.0)
Albumin, g/dL *	4.0 (3.7–4.3)
Prothrombin time, % *	87.9 (81.5–97.0)
Child–Pugh score, 5:6	326:103
Creatinine, mg/dL *	0.79 (0.67–0.94)
CRP, mg/dL *	0.09 (0.04–0.25)
CRP ≤ 1.0 mg/dL, *n* (%) *	37 (8.6)
ALBI score *	−2.68 (−2.39 to −2.94)
mALBI grade, 1:2a:2b	249:115:65
ICG-R15 (%) *	14.4 (9.3–20.9)
Positive for esophageal varices, *n* (%)	56 (13.1%)
FIB-4 index *	3.06 (2.06–4.24)
AFP, ng/mL *	9.4 (3.9–56.9)
Single tumor, *n* (%)	340 (79.2)
Maximum tumor size, cm *	3.5 (2.4–5.5)
Open:laparoscopic	304:125
Hepatectomy, non-anatomic:segmentectomy:sectionectomy:hemitectomy	141:61:130:97
Microvascular invasion present, *n* (%)	297 (69.2)
Operation time, minutes *	323 (256–409)
Blood loss, mL *	858 (257–1121)
Observation period, months *	35 (15–68)
GPS, 0:1:2	357:63:9
Neo-GPS, 0:1:2	239:164:26
Clavien–Dindo classification ≥ 3, *n* (%)	83 (19.3)
Cause of death, (liver failure:HCC:resection complication:other:unknown)	15:85:8:27:9

* Median. Values in parentheses show interquartile range, unless otherwise indicated. HCV: hepatitis C virus; HBV: hepatitis B virus; CRP: C-reactive protein; ALBI score: albumin–bilirubin score; mALBI grade: modified ALBI grade; ICG-R15: indocyanine green retention rate at 15 min; FIB-4 index: fibrosis-4 index; AFP: alpha-fetoprotein; GPS: Glasgow prognostic score; HCC: hepatocellular carcinoma.

**Table 2 cancers-14-01402-t002:** Clinical features of patients in open and laparoscopic hepatectomy groups.

	Open Hepatectomy (*n* = 304)	Laparoscopic Hepatectomy (*n* = 125)	*p*-Value
Age, years *	73 (67–78)	73 (65–80)	0.772
Gender, male:female	238:66	88:37	0.105
Body mass index, kg/m^2^ *	23.1 (20.8–25.5)	23.5 (20.926.0)	0.261
Etiology, HCV:HBV:HBV&HCV:alcohol:others	106:54:5:33:106	48:20:0:20:37	0.399
Positive for diabetes mellitus, *n* (%)	103 (33.9)	38 (30.4)	0.500
Aspartate transaminase, U/L *	35 (25–50)	31 (25–46)	0.356
Alanine aminotransferase, U/L *	29 (18–46)	24 (19–41)	0.260
Platelets, 10^4^/µL *	16.3 (12.4–21.2)	14.7 (11.018.6)	0.006
Total bilirubin, mg/dL *	0.7 (0.6–1.0)	0.8 (0.7–1.0)	0.007
Albumin, g/dL *	4.0 (3.7–4.3)	4.1 (3.7–4.4)	0.266
Prothrombin time (%) *	88.5 (81.8–88.1)	87.0 (78.9–93.9)	0.016
Child–Pugh score, 5:6	237:67	89:36	0.138
Creatinine, mg/dL *	0.79 (0.67–0.94)	0.80 (0.68–0.94)	0.549
CRP, mg/dL *	0.11 (0.04–0.31)	0.07 (0.04–0.15)	0.004
CRP ≤1.0 mg/dL, *n* (%) *	34 (11.2)	3 (2.4)	0.002
ALBI score *	−2.65 (−2.41 to −2.91)	−2.70 (−2.36 to −2.99)	0.590
mALBI grade, 1:2a:2b	177:84:43	72:31:22	0.613
ICG-R15 (%) *	13.8 (9.1–19.2)	16.4 (10.7–25.8)	0.002
Positive for esophageal varices, *n* (%)	35 (11.5)	21 (16.8)	0.113
FIB-4 index *	3.04 (2.01–4.14)	3.09 (2.29–4.62)	0.221
AFP, ng/mL *	9.5 (3.9–69.6)	9.4 (3.9–45.0)	0.773
Single tumor, *n* (%)	227 (74.7)	113 (90.4)	<0.001
Maximum tumor size, cm *	4.0 (2.5–6.6)	2.7 (2.0–3.5)	<0.001
Hepatectomy, non-anatomic:segmentectomy:sectionectomy:hemitectomy	87:33:103:81	54:28:27:16	<0.001
Microvascular invasion present, *n* (%)	218 (71.7)	79 (63.2)	0.086
Operation time, minutes *	333 (262–407)	298 (239–416)	0.253
Blood loss, mL *	750 (458–1337)	167 (70.0–341)	<0.001
Observation period, months *	40 (20–71)	24 (9–53)	<0.001
GPS, 0:1 or more	251:53	106:19	0.670
Neo-GPS, 0:1 or more	168:136	71:54	0.831
Clavien–Dindo classification ≥ 3, *n* (%)	69 (22.7)	14 (11.2)	0.007
Cause of death, liver failure:HCC:resection complication:other:unknown	9:66:7:20:6	6:19:1:7:3	0.570

* Median. Values in parentheses show interquartile range, unless otherwise indicated. HCV: hepatitis C virus; HBV: hepatitis B virus; CRP: C-reactive protein; ALBI score: albumin–bilirubin score; mALBI grade: modified ALBI grade; ICG-R15: indocyanine green retention rate at 15 min; FIB-4 index: fibrosis-4 index; AFP: alpha-fetoprotein; GPS: Glasgow prognostic score; HCC: hepatocellular carcinoma.

**Table 3 cancers-14-01402-t003:** Correlation between Clavien–Dindo classifications and elevated nutritional scoring.

	Clavien–Dindo Classification ≤ 2 (*n* = 346)	Clavien–Dindo Classification ≥ 3 (*n* = 83)	*p*-Value
GPS 0	301 (87.0%)	56 (67.5%)	
GPS ≥ 1	45 (13.0%)	27 (32.5%)	<0.001
Neo-GPS 0	210 (60.8%)	29 (34.9%)	
Neo-GPS ≥ 1	136 (39.2%)	54 (65.1%)	<0.001

GPS: Glasgow prognostic score.

## Data Availability

Please refer to suggested Data Availability Statements at https://www.mdpi.com/journal/cancers/instructions#suppmaterials (accessed on 14 February 2022). Due to the nature of this research, participants in this study could not be contacted regarding whether the findings could be shared publicly, thus supporting data are not available. The datasets generated and/or analyzed for the current study are not publicly available due to the nature of the research, as noted above.

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
