# Peer review of "Predicting Complications following Surgical Resection of Hepatocellular Carcinoma Using Newly Developed Neo-Glasgow Prognostic Score with ALBI Grade: Comparison of Open and Laparoscopic Surgery Cases"

_cancers, 2022, doi:10.3390/cancers14061402_

Round 1

Reviewer 1 Report

The Authors reported on a series of 429 patients affected with hepatocellular carcinoma who were operated on with liver resection on negative surgical margins between 2010 and 2020. They investigated the prognostic role of a modified Glasgow prognostic score, based on ALBI grade 1 instead of albumin level, to assess the nutritional status of such patients. The manuscript if of interest, as the topic is relevant, and methodology is clear and robust.

  • In the present study, an improved GPS scoring method (neo-GPS), based on ALBI
    grade is used. I would suggest the autgors to better explain the rationale for this marker to be used and the reason why it is assumed to be a more sensitive outcome prognosticator.
  • Would you provide also details on the reason why the grade 1 cut-off has been used?
  • Amongst the limitation, I would add the lack of an external validation.

Author Response

Replies to comments of Rev. 1.

The Authors reported on a series of 429 patients affected with hepatocellular carcinoma who were operated on with liver resection on negative surgical margins between 2010 and 2020. They investigated the prognostic role of a modified Glasgow prognostic score, based on ALBI grade 1 instead of albumin level, to assess the nutritional status of such patients. The manuscript if of interest, as the topic is relevant, and methodology is clear and robust.

  • In the present study, an improved GPS scoring method (neo-GPS), based on ALBI
    grade is used. I would suggest the autgors to better explain the rationale for this marker to be used and the reason why it is assumed to be a more sensitive outcome prognosticator. 
  • Would you provide also details on the reason why the grade 1 cut-off has been used?

Reply to above 2 comments:

Thank you for your useful comments, we added sentences into Discussion as follows:

“In fact, although the cut-off value of albumin in GPS is 3.5 g/dL, even the exemplar patient with a very low total bilirubin level of 0.3 mg/dL and an albumin level of 3.5 g/dL is categorized as the middle grade of ALBI (grade 2, score -2.51). On the other hand, it has been reported that ALBI grade 1 is superior to Child-Pugh A in prediction of prognosis of HCC patients [14]. These facts suggest that the cut-off value of albumin of GPS (3.5 g/dL) is an unsatisfactory criterion for HCC patients whose basal diseases are often liver cirrhosis, and that ALBI grade should be used instead of albumin (3.5g/dL) as a nutritional factor. Compared to GPS, which was developed as a nutritional assessment method for patients with cancers other than HCC, ï½”he present newly developed neo-GPS using ALBI grade 1 might play a greater role for predicting prognosis and fit for HCC patients with chronic liver disease than GPS, because ALBI grade 1 can indirectly assess the nutritional status for the boundaries of amino acid imbalance in patients with chronic liver disease [9].”

  • Amongst the limitation, I would add the lack of an external validation.

Reply:

Thank you for your advice. We revised “limitation parts” in Discussion as follows:

“This study is limited by its retrospective nature. Furthermore, tumor number and size were greater in the open hepatectomy as compared to the laparoscopic hepatectomy cases. Because of lacking data (e.g. neutrophils and lymphocyte), we could not perform comparison among the present neo-GPS and neutrophil/lymphocyte ratio or nutritional index, a future study to compare among the present neo-GPS and other nutritional assessment tools should be planed. Moreover, it is considered that a randomized control trial for obtaining more concrete conclusions is necessary in addition to validation studies, while accumulation of greater numbers of patients and a longer observation period will be useful for presenting more definitive conclusions.”

Reviewer 2 Report

In this submission, Kaibori et al. examined the usefulness of a prognostic predictive nutritional assessment tool for hepatocellular carcinoma (HCC) patients treated with surgical resection. This is an interesting study but a few mechanisms are not clear in its current format which I have listed below. I suggest this manuscript to be accepted with subject to major revisions.

  • Introduction - needs to be extended in terms of nutritional assessment and the utility of other state-of-the-art techniques to validate these markers.
  • The varying numbers of n in table 2 should clearly be explained to readers. same applies to table 3.
  • Discussion lacks the in-depth and mechanistic discussion around the technology, and its comparison with other modalities.
  • There is no mention of challenges in this technology, and what are the advantages offered by this approach?
  • what are the translational problems?

Author Response

In this submission, Kaibori et al. examined the usefulness of a prognostic predictive nutritional assessment tool for hepatocellular carcinoma (HCC) patients treated with surgical resection. This is an interesting study but a few mechanisms are not clear in its current format which I have listed below. I suggest this manuscript to be accepted with subject to major revisions.

  • Introduction - needs to be extended in terms of nutritional assessment and the utility of other state-of-the-art techniques to validate these markers.

Reply:

Thank you for your useful suggestion. We revised Introduction as follows:

“In addition, as compared to patients with a low GPS score (<1), those with a score ≥1 have been reported to have a higher rate of complications following surgical resection for advanced gastric cancer (16% vs. 31%, P=0.022) [8]. On the other hand, because HCC often develops in patients with chronic liver disease, mainly with liver cirrhosis, development of a new nutritional evaluation method, which fits for chronic liver disease patients, is needed.

In the present study, an improved GPS scoring method (neo-GPS) using albumin-bilirubin (ALBI) grade 1 reported to indicate the approximate borderline of amino acid imbalance [9] instead of serum albumin, was developed and a prognostic predictive value of newly developed neo-GPS was evaluated in comparing with GPS score following surgical resection.”

  • The varying numbers of n in table 2 should clearly be explained to readers. same applies to table 3.

Reply:

Thank you for your pointing out our miscount and mis-writings. We apologize to mis-wrote these results. We checked database and re-counted them.

All number and related rates were revised. Revised parts were written with red-colored letters.

  • Discussion lacks the in-depth and mechanistic discussion around the technology, and its comparison with other modalities.
  • There is no mention of challenges in this technology, and what are the advantages offered by this approach?

Replies to above 2 comments:

Thank you for your important advice.

We added and revised sentences as follows: “In fact, although the cut-off value of albumin in GPS is 3.5 g/dL, even the exemplar patient with a very low total bilirubin level of 0.3 mg/dL and an albumin level of 3.5 g/dL is categorized as the middle grade of ALBI (grade 2, score -2.51). On the other hand, it has been reported that ALBI grade 1 is superior to Child-Pugh A in prediction of prognosis of HCC patients [14]. These facts suggest that the cut-off value of albumin of GPS (3.5 g/dL) is an unsatisfactory criterion for HCC patients whose basal diseases are often liver cirrhosis, and that ALBI grade should be used instead of albumin (3.5g/dL) as a nutritional factor. Compared to GPS, which was developed as a nutritional assessment method for patients with cancers other than HCC, ï½”he present newly developed neo-GPS using ALBI grade 1 might play a greater role for predicting prognosis and fit for HCC patients with chronic liver disease than GPS, because ALBI grade 1 can indirectly assess the nutritional status for the boundaries of amino acid imbalance in patients with chronic liver disease [9].” in Discussion, and “This study is limited by its retrospective nature. Furthermore, tumor number and size were greater in the open hepatectomy as compared to the laparoscopic hepatectomy cases. Because of lacking data (e.g. neutrophils and lymphocyte), we could not perform comparison among the present neo-GPS and neutrophil/lymphocyte ratio or nutritional index, a future study to compare among the present neo-GPS and other nutritional assessment tools should be planed. Moreover, it is considered that a randomized control trial for obtaining more concrete conclusions is necessary in addition to validation studies, while accumulation of greater numbers of patients and a longer observation period will be useful for presenting more definitive conclusions.” in the Limitation parts of Discussion.

  • what are the translational problems?

Reply:

Thank you for your comment.

We ordered native checking our manuscript to “Intermed English Services (by Mr. Mark Benton)” (www.intermed-jp.com). If you feel re-ordering native-checking, we order it again.

Round 2

Reviewer 2 Report

I am pleased to recommend the revised manuscript for publication in Cancers.